# Medical and Psychological Aspects of Pregnancy in Women with Obesity and after Bariatric Surgery

**DOI:** 10.3390/nu15194289

**Published:** 2023-10-08

**Authors:** Monika Bąk-Sosnowska, Beata Naworska

**Affiliations:** 1Center for Psychosomatics and Preventive Healthcare, WSB University in Dabrowa Gornicza, 41-300 Dabrowa Gornicza, Poland; 2Department of Gynaecology and Obstetrics, Faculty of Health Sciences in Katowice, Medical University of Silesia in Katowice, 40-751 Katowice, Poland; bnaworska@sum.edu.pl

**Keywords:** maternal obesity, bariatric surgery, pregnancy, psychological aspects, obstetric outcome, perinatal outcome

## Abstract

Chronic diseases are potential risk factors for pregnancy duration and neonatal outcomes. This narrative review aimed to summarize the research results on the specifics of pregnancy in women with obesity and after bariatric surgery. PubMed and Google Scholar databases were searched. Systematic reviews, meta-analyses, clinical trials, and references to identified articles from the last ten years (2013–2023) were included. Ultimately, 107 literature items were qualified. It has been shown that women with obesity planning pregnancy should reduce their body weight because obesity is a risk factor for adverse obstetric and neonatal outcomes. Bariatric surgery effectively reduces excessive body weight and the health risks in women with obesity during pregnancy and their offspring. However, at least a year interval between surgery and conception is recommended. An interdisciplinary medical team should provide patient care during pregnancy with knowledge and skills related to people after bariatric surgery. Due to the increased risk of mental disorders, especially depression, it is necessary to constantly monitor the mental state of women and provide psychological support and education on a healthy lifestyle during pregnancy and the postpartum period.

## 1. Introduction

Obesity is a complex chronic disease in which abnormal or excessive adipose tissue accumulation worsens health, increases the risk of long-term medical complications, and shortens life expectancy. The most common health consequences of obesity are type 2 diabetes, hypercholesterolemia, cardiovascular diseases, cancer, asthma, sleep apnea, osteoarthritis, low quality of life, and depression. Obesity in adults is often classified based on a BMI (weight (kg)/height (m)^2^) equal to or greater than 30. Other helpful measures include waist circumference, waist-to-hip ratio, midsection arm, bioelectrical impedance measurements of body fat percentage and mass, and direct imaging methods like computed tomography or magnetic resonance imaging (alternatively ultrasonography) of subcutaneous and visceral fat thickness. In addition, when diagnosing obesity, it is worth considering other criteria, such as psychological, social, and economic [1].

The prevalence of obesity in developed countries is steadily increasing. In the United States, 41.9% of adults and 39.6% of women aged 20–39 suffer from this disease. Among women in this group, 12.4% are severely obese (BMI ≥ 40 kg/m^2^) [2]. The problem also affects women during pregnancy. European data indicate that 7–35% of mothers have a BMI > 30 at the beginning of pregnancy, and the differences in results are primarily related to social and educational inequalities [3]. Not only obesity before pregnancy but also excessive weight gain during pregnancy is detrimental to the mother’s and child’s health. It can be caused by many factors: genetic, sociodemographic, environmental, socio-economic, psychological, and medical [4,5]. Despite the recognized benefits of a healthy diet and physical activity during pregnancy, a low percentage of women adhere to prenatal nutritional and physical activity guidelines [6,7]. This situation contributes to postpartum weight retention, which strongly predicts obesity later in life [8].

Due to the increasing risk of obesity and its complications in women of reproductive age and their children, it is necessary to seek therapeutic interventions to ensure the excellent effectiveness of obesity treatment. Although the basis is a healthy lifestyle change, bariatric surgery plays an important role, especially in the case of morbid obesity. Its advantage is a significant loss of body weight in a relatively short time, which allows one to achieve positive health effects related to the quality of life [9]. Regardless of the method of treating obesity, it is worth bearing in mind that both during pregnancy and in other periods of life, the psychosocial context is also crucial for human health. Psychological factors underlying the development, course, and treatment of somatic diseases allow us to understand their nature better and increase effectiveness in clinical practice through comprehensive interaction.

This narrative review first aims to show the specifics of pregnancy in women with obesity. We consider preconception conditions, medical and psychological aspects of pregnancy, possible childbirth complications, and the development of children of women with obesity during pregnancy. Next, we present issues related to reducing excessive body weight in women of reproductive age, particularly the medical and psychological benefits and risks associated with bariatric surgery. Next, we discuss the impact of bariatric surgery on the course of pregnancy and obstetric outcomes. In this case, we also treat the subject comprehensively, considering medical and psychological issues. Finally, we present the key findings related to women’s pregnancy after bariatric surgery and the resulting practical recommendations, and we propose prospects for further research in this area.

## 2. Methods

The methodology of the presented narrative review was based on PubMed and Google Scholar searches. The search was carried out in August 2023 using the following keywords: female obesity, pregnancy after bariatric surgery, obstetric outcomes, neonatal outcomes, and psychological aspects of obesity. A detailed list of keywords is included in Table 1.

The exclusion criteria were studies published before 2013, articles not focusing on medical and psychological factors of pregnancy with obesity and after bariatric surgery, as well as textbooks, letters, editorials, conference abstracts, and commentaries. The literature items included in this review were systematic reviews, meta-analyses, clinical trials, and references to identified articles. The bibliographic lists of each article were searched manually. The authors selected all studies in English with the full text available in human studies, with particular emphasis on the impact of obesity in pregnancy on the biological and mental health of the mother and child. In addition, the WHO and the American College of Obstetricians and Gynecologists’ websites were searched for recommendations regarding pregnancy in women with obesity. This narrative review summarizes the evidence from the last ten years (2013–2023).

Ultimately, 107 thematically coherent literature items relevant to the achievement of the assumed goal were qualified for the review, comprising 103 peer-reviewed scientific articles, a report concerning the WHO guidelines, and three articles involving recommendations of scientific associations regarding pregnancy in women with obesity.

## 3. Specificity of Women with Obesity during Pregnancy

Obesity during pregnancy harms the health of both the mother and her baby. In studies of adverse pregnancy outcomes in women with obesity, repeated predictors include pre-pregnancy type 1 diabetes, non-Caucasian ethnicity, specific age (<20 and ≥35 years), abdominal obesity, and history of bariatric surgery [10]. The problems most frequently indicated in the reviews of scientific literature and meta-analyses of the negative health consequences of gestational obesity for mothers include gestational diabetes mellitus (GDM) [10,11,12,13,14,15], gestational hypertension (GH) [11,12,13,14,15,16]), pre-eclampsia (PE) [10,11,13,15,16], peripartum cardiomyopathy (PPCM) [16], obstructive sleep apnea (OSA) [16], vitamin D deficiency [16], and other medical problems (thromboembolism, stroke, myocardial infarction, and wound infections after surgery) [12]. The risk of GH and PE is 3–10 times higher in women with obesity than in women with normal body weight and the highest in women with severe obesity who gained much weight during pregnancy [16].

In high-income countries, 1 in 10 mothers experience perinatal mental health problems. These include primarily prenatal anxiety and perinatal depression [17]. Women with excessive body weight before pregnancy, regardless of the amount of weight gained during pregnancy, are more likely to suffer from postpartum depression than their peers with normal body weight [18]. Obesity during pregnancy is associated with poor mental health, including anxiety, depression, and postpartum depression, and it also increases the risk of binge eating disorders and severe mental illness [19]. The relationship between depression and obesity is complex and bidirectional. Women during pregnancy with depression are more likely to suffer from obesity than women without depression [20,21]. Similarly, women with obesity are more likely to experience depressive symptoms during pregnancy and postpartum than women with normal body weight [21,22].

Maternal obesity also increases the risk of health problems for the baby. They concern the fetal period, childbirth, and later development. Based on reviews and meta-analyses, the most common complications include spontaneous abortion (SAB), as well as recurrent abortion [12,16]; fetal defects (neural tube, heart, craniofacial, limbs, and hydrocephalus) [11,16]; preterm birth [10,11,13,16]; complicated and prolonged delivery [12]; cesarean delivery (CD) [11,12,13,15]; stillbirth [10,13,16]; postpartum complications (increased risk of postpartum hemorrhage; the occurrence of venous thromboembolism); complications from prolonged immobilization after cesarean section (skin breakdown, cardiac deconditioning, deep vein thrombosis, muscle atrophy, urinary stasis, constipation, pain management problems, pulmonary complications, and depression); reduced frequency and duration of breastfeeding [16]; low birth weight (LBW) [10,13,14] or fetal macrosomia (FM) (>4000 g at birth) [14,15,16,23]; and small for gestational age (SGA) [10,12,13] or large for gestational age (LGA) [11,16]. Inconsistent reports of fetal size concerning gestational age and birth weight can be attributed to various conditions during and before pregnancy. Clinical practice shows that, among other things, a woman’s body weight before pregnancy and maternal weight gain during pregnancy affect the condition and parameters of the fetus. Nutritional deficiencies that are common in people suffering from obesity can cause insufficient nutrition of the fetus, which results in its slower growth and lower birth weight. In addition, reduced levels of maternal adiponectin in women with obesity may reduce the transport of amino acids across the placenta and inhibit fetal growth [24].

Based on 25 cohort studies from Europe and North America, the risk of pregnancy complications was estimated depending on the baseline BMI and weight gain during pregnancy [13]. At least one of the following adverse outcomes was considered: pre-eclampsia, gestational hypertension, gestational diabetes, cesarean delivery, preterm delivery, and small or large size for gestational age at birth. They found that the lowest risk was 26.7% for women with a BMI of less than 18.0 and a pregnancy weight gain of 26.0 kg to 27.9 kg. The most significant risk was 94.4% for women with a BMI of 40.0 or greater and a gestational weight gain of 20.0 kg to 21.9 kg. In women with grade 3 obesity, any adverse pregnancy outcome occurred in 61.1% of cases. On this basis, the optimal ranges of weight change during pregnancy were defined. They are presented in Table 2. 

Maternal pre-pregnancy obesity and excessive gestational weight gain have not only short-term consequences for the offspring but also increase the risk of health problems in the future. Children of mothers with obesity have an increased risk of overweight and obesity, ischemic heart disease, stroke, type 2 diabetes, and asthma. Preliminary evidence suggests potential implications for immune and infectious disease outcomes. Maternal obesity may additionally lead to poorer cognitive functions in offspring and an increased risk of neurodevelopmental disorders, including cerebral palsy. The mechanisms underlying these dependencies are complex. They include the interplay between evolving metabolic and immune responses, as well as maternal nutrition, adiposity, and diversity of gut microbiota [25].

The intrauterine environment plays a mediating role in fetal programming due to the mother’s somatic condition and her mental health. Pregnancy depression increases the risk of future emotional and behavioral problems in the offspring, including depression, anxiety, and problems with concentration. Because depression and obesity often co-occur, pregnant women may experience both disorders, increasing the risk of potential adverse effects on their offspring. Studies assessing the impact of the simultaneous occurrence of obesity and depression in pregnant women on long-term outcomes in offspring confirm that they increase the risk of suboptimal physical, cognitive, and socioemotional development; poorer academic performance; physical problems; and mental disorders in later life [21].

Epidemiological data indicate, however, that about 40–50% of women with obesity gain more weight during pregnancy than is recommended [26], which contributes to negative health consequences similar to pre-pregnancy obesity. The causes of excessive gestational weight gain (GWG) are highly individualized, but based on a Chinese cohort study of over 3000 women, the most important are age, education, race, and region of residence. In the cited study, body weight was better controlled by women aged 20–25 than younger or older women, as well as women with education other than lower or university; women not belonging to ethnic minorities; and those living in the south of the country, which is associated with specific cultural conditions in nutrition and climate [14].

Interesting results are also provided by a meta-analysis of the psychological causes of excessive gestational weight gain (GWG). It considers three psychological constructs: affect, beliefs related to weight gain and eating behavior, and personality. The risk factors for GWG were higher dietary restrictions, perceived barriers to healthy eating, negative attitude toward weight gain, negative body image, fear of gaining weight, high target weight gain, wrong perception of own body weight, and less knowledge about weight gain. On the other hand, protective factors concerned higher self-efficacy regarding healthy eating, lower than recommended target weight gain, and internal locus of control for weight gain [27]. Although in the quoted meta-analysis, the effect was not associated with excessive GWG, a review of prospective cohort studies provides different results, which show that perinatal depression and anxiety are predictors not only of excessive GWG but also of postpartum weight maintenance (PPWR) [28]. Another important psychological factor is the sense of coherence. It is a personality trait that indicates the ability to understand, manage, and give meaning to situations; increases stress resistance; and promotes human development. It was shown that the sense of coherence at the beginning of the next pregnancy was independently related to women’s BMI and body fat percentage before pregnancy. Body weight and body fat percentage at the beginning of the subsequent pregnancy appear to be significantly related to a person’s ability to cope with stressful everyday events [29].

A review of commentaries of women during pregnancy and the postpartum period from high-income countries evaluating participants’ experiences of weight management during pregnancy showed that the following are essential [30]:-Awareness and beliefs about gaining weight and controlling body weight—many women during pregnancy were characterized by a lack of knowledge about the risks associated with overweight and obesity during pregnancy and a lack of awareness of recommendations regarding physical activity and diet;-Social and environmental impact—discussions with well-wishers about weight and access to information sources were positive; in turn, the negative factor is the social stigma associated with obesity in general and especially during pregnancy; this results in tabooing the topic and reduces the chance for constructive discussions and actions in this area; an additional factor is also quick and easy access to cheap fast-food food and the lack of affordable exercise facilities;-Antenatal care—women who had negative interactions with health professionals felt ashamed and stigmatized; this made them feel a barrier to discussing topics related to body weight, as well as to engaging in activities related to overweight reduction; moreover, the subjects reported that the issue of weight was not treated as a priority and often not given enough attention, and advice was rarely given, superficially, and hastily.

A systematic review of current data on antenatal care for women with obesity revealed that healthcare professionals are too often unsure how to talk to women and instead avoid or cover essential topics related to obesity. As a result, women with obesity often do not receive advice on recommended weight gain, nutrition, or physical activity during pregnancy and feel stigmatized because of obesity. The opposite is the partnership attitude, which facilitates understanding the patient’s situation and motivation [11]. In addition, people-first language, i.e., focusing on the person and not the problem/disease, is exemplified, among others, in the use of the term “person with obesity” instead of “obese person” [11,31]. The person-centered approach and motivational interviewing are also effective methods to reduce the symptoms of depression during pregnancy and adverse pregnancy outcomes [32].

The need for multispecialty care for a woman during pregnancy and in labor who suffers from obesity at the same time results from health complications of obesity but also from special medical needs related to excessive body weight. These include appropriate logistics of the medical facility (wider doors, lifts with increased permissible weight); medical equipment (sleeve width for blood pressure measuring devices, surgical beds or tables with adequate load capacity, portable or ceiling-mounted lifting equipment, bariatric wheelchairs); medical procedures (increased drug doses, ultrasound assessment of fetal presentation due to the lower reliability of transvaginal assessment, inclusion of respiratory therapy or the use of incentive spirometry); and medical personnel (specialist knowledge and skills related to, e.g., positioning, carrying patient, venipuncture or intubation) [16].

## 4. Reduction in Excessive Body Weight in Women of Reproductive Age

In women with obesity, a slight increase in body weight during pregnancy is recommended, and even losing weight is considered beneficial [13]. Studies confirm that the probability of having a child large for gestational age (LGA) and macrosomia, as well as the risk of maternal complications, such as cesarean section delivery, are significantly lower in women with obesity and gestational weight loss (GWL) [33].

Exercise during pregnancy is one strategy for reducing weight gain and excessive body mass. It is recommended by most professional organizations in their guidelines for caring for women with obesity during pregnancy. Activities such as swimming, brisk walking, or strength training are safe, but their goal should be to stay in shape rather than to achieve sports results [11]. A comparison of prenatal physical activity guidelines from many high-income countries shows that most recommend seeking medical advice before starting or continuing an exercise program or moderate physical activity and avoiding sports that involve a risk of falls, injury, or collision [34]. Many reports confirm the health benefits of physical activity during pregnancy, such as lower gestational weight gain, reduced insulin resistance, lower risk of gestational diabetes, and appropriate birth weight of the newborn [35,36]. However, other studies do not confirm that women with obesity who are physically active during pregnancy differ from women with obesity covered by standard care in terms of such parameters as average gestational age at delivery, cesarean section frequency, birth weight of the newborn, body size of newborns, skinfold thickness, placental mass ratio, Apgar score, mean length of hospital stay [37]. Differences were not noted, probably because study participants adhered poorly to activity recommendations.

Regarding nutrition, studies have shown disappointing results when testing the effects of dietary interventions on gestational weight gain, pregnancy outcomes, and birth outcomes in women with obesity. No differences were found whether women were offered weekly coaching on an adapted DASH diet [38]; serial weighing and dietary advice [39]; or a series of in-person counseling sessions on weight, exercise, and diet during pregnancy [40].

Clinical practice shows that the pregnancy period can redirect a woman’s attention to the child’s needs. If she does not have sufficient education, she may consider physical activity unimportant or even threaten the child. In addition, pregnancy can be associated with strong, negative emotions and even mental disorders, making it difficult for a woman to focus on a healthy lifestyle and taking care of her weight. These are additional considerations for starting obesity treatment before the planned pregnancy.

Reducing excessive body weight before pregnancy is critical, and at the same time, unhealthy weight loss before conception can have adverse consequences for the mother and her offspring. For this reason, the most essential issue is a healthy diet. It means food and drinks rich in nutrients, which is associated with the consumption of an appropriate serving of vegetables, especially legumes, fruits, whole grains, low-fat dairy products, lean meat, seafood, nuts, liquid oils, and the preference for low-sodium products, low-fat saturated fats, and sugars [41]. A healthy diet for women is essential before conception to prepare for the increased nutrient requirements of early pregnancy, reach the recommended nutritional goals and weight gain levels in the following months of pregnancy, and increase the chance of successful lactation and return to pre-pregnancy weight after childbirth [42]. The child’s neurodevelopmental processes in the first 1000 days after conception also require essential nutrients such as protein, long-chain polyunsaturated fatty acids, zinc, copper, iodine, iron, folic acid, and choline. At all stages of life, including pregnancy, nutrients often deficient are vital: calcium, vitamin D, potassium, and dietary fiber [43].

Changing the lifestyle to a pro-health lifestyle has a beneficial effect on body weight reduction and reproductive outcomes, such as an increase in the frequency of clinical pregnancies, live births, and natural conceptions [44]. Conservative treatment of obesity includes behavioral (diet, physical activity, and sleep); psychological (planning strategies for weight reduction, stress reduction, and emotional self-regulation); and pharmacological interventions [45]. In many cases, however, the effectiveness of conservative treatment is unsatisfactory, which encourages the consideration of surgical intervention in treating obesity.

Bariatric surgery (BS) is an invasive procedure that reduces the stomach’s capacity, impairs the absorption of nutrients, or a combination of both. Recent scientific reports confirm that it also results in specific changes in hormonal balance, so we are currently discussing metabolic and bariatric surgery (MBS). The most commonly used treatments include sleeve gastrectomy (SG), Roux-en-Y gastric bypass (RYGB), and adjustable gastric band (AGB). In SG, the greater curvature of the stomach is resected, reducing stomach volume by 75%, thus limiting food intake. Ghrelin-producing secreting endocrine cells present in the greater curvature of the stomach are also reduced, which is positive in appetite reduction. RYGB is a mixed procedure in which the stomach volume is reduced to approximately 15 to 30 mL, and the absorption of nutrients is impaired by bypassing part of the small intestine and diverting the food flow to the distant small intestine. It influences limited oral intake and malabsorption. Additionally, increased gut hormone secretion (including GLP-1 and PYY) hormones may diminish appetite and result in better glucose homeostasis. In AGB, an inflatable restrictive band is placed around the stomach’s upper portion, creating a small pouch with a narrow opening to the lower stomach. It is adjusted by adding or removing fluid to the band via a subdermal port, which reduces stomach capacity and appetite [46]. Scientific discoveries regarding the metabolic mechanisms underlying the development of obesity have resulted in the use of the term metabolic and bariatric surgery (MBS), and the dominant procedures are sleeve gastrectomy and RYGB. MBS is recommended for individuals with a BMI > 35 kg/m^2^ (regardless of presence, absence, or severity of comorbidities) and taken under consideration for individuals with a BMI of 30–34.9 kg/m^2^ and metabolic disease. In the Asian population, a BMI > 25 kg/m^2^ should indicate clinical obesity, and MBS should be offered for individuals with a BMI > 27.5 kg/m^2^ [47].

Literature reviews and meta-analyses of studies indicate numerous health benefits of treating obesity with bariatric surgery. Beneficial effects have been observed in cancer incidence, mortality, cardiovascular risk, polycystic ovary syndrome (PCOS), risk of kidney stones, albuminuria, urinary incontinence, fecal incontinence, Barrett’s esophagus, and diabetic retinopathy [9]. There is a significant improvement in pelvic floor disorders, including urinary incontinence, pelvic organ prolapse, and colorectal symptoms [48]. Long-term maintenance of postoperative weight reduction contributes to better control of diabetes, physical activity, and reduction in antidiabetic and hypotensive drugs [49]. The surgical treatment of obesity improves the overall quality of life (QoL) [50,51,52] and health-related quality of life (HrQoL) [53]. The mental state and emotional regulation improve, and the symptoms of depression and anxiety disorders decrease [9,53].

More than 50% of bariatric surgeries are performed in women of reproductive age [54], and thus the benefits in this group of patients are worth emphasizing. Studies indicate a reduction in somatic symptoms, improvement in general health and well-being, memory improvement, increased self-control, mood improvement, anxiety and fear reduction [55], lower depression scores, improved body image, increased self-esteem, feeling more comfortable with oneself and feeling worthy of love [56], a sense of solving an unbearable problem, learning new limits of one’s abilities, and hope for normalization [57]. Patients operated on due to obesity also emphasize a better quality of sex life in desire, arousal, lubrication, orgasm, satisfaction and pain, depressive symptoms, and self-esteem [55,56,58]. In addition, better sexual performance [48], more sexual behavior, higher sense of sexual attractiveness [55], and think about being a parent [56] have also been reported.

It is worth emphasizing that in most reports on psychological and somatic benefits after bariatric surgery, no information was provided on whether patients had undergone abdominoplasty. Therefore, its role as an intermediary in improving psychophysical health cannot be analyzed. Meanwhile, for some patients, significant weight loss after bariatric surgery means burdensome excess skin, mainly on the abdomen and the thighs, arms, breasts, back, and face. Excess skin can result in intertrigo, skin infections, mobility problems, negative body image, depression, and social dysfunctions [59]. One study has shown that abdominoplasty is associated with reduced secondary weight gain after RYGB. However, it remains unclear whether this is due to increased bodily satisfaction and better physical performance or a biological response to body fat reduction [60].

Despite several positive health consequences of bariatric surgery, it should also be noted that they may be associated with undesirable effects. Typical are nausea, vomiting, reflux, or dumping syndrome. They impair nutrient intake and cause vitamin and mineral deficiencies [61]. Research reviews confirm that many patients continue to eat low-quality diets after bariatric surgery [62]. These facts show the importance of health education before and after surgery. It should be an inseparable element of obesity treatment, including surgical treatment. Various somatic symptoms (e.g., abdominal pain, fatigue, anemia, etc.) and hospitalizations may occur more frequently after bariatric surgery [63]. There are also reports of negative effects on mental state, including an increased risk of self-harm [64], suicide attempts, and suicides [64,65]. It seems that the psychological condition of patients after bariatric surgery is of crucial importance for the results of psychological functioning of patients after bariatric surgery, and thus the correct qualification for surgery. Psychological assessment should be an inseparable element of the qualification, and it is recommended by the guidelines of the European Association for the Study of Obesity (EASO). Each current acute psychopathology is a contraindication to bariatric surgery [45].

## 5. The Influence of Bariatric Surgery on Pregnancy and Neonatal Outcomes

In women of reproductive age, contraception is an essential postbariatric recommendation. The intrauterine device is recommended as first-line therapy because the effectiveness of oral contraceptive pills may be reduced due to malabsorption [46,66]. The results of scientific studies confirm the lack of differences in weight loss in the first year after surgery between women using different methods of contraception. At the same time, they indicate that despite the recommendation to avoid pregnancy in the first period after surgery, as many as one third of patients do not use any form of contraception [67].

The recommendation to use contraception in the postoperative period is associated with a potential risk to the child and the pregnant woman herself. In clinical practice, a cautious waiting period of at least two years after bariatric surgery to become pregnant is often suggested. This is because the typical duration of the initial period of significant and rapid weight loss and increased potential risk of nutritional deficiencies is approximately 12–24 months [68]. The guidelines recommend multivitamin and mineral supplementations before conception and throughout pregnancy. In particular, a biochemical assessment is strongly recommended to determine specific micronutrient requirements. The supplement should contain at least copper (2 mg), zinc (8–15 mg per 1 mg copper), calcium (1200–1500 mg in divided doses), selenium (50 μg), folic acid (0.4–1 mg, 4–5 mg if obesity or diabetes), iron (45–60 mg or >18 mg after AGB, thiamine (>12 mg), vitamin D (>40 mcg), vitamin E (15 mg), vitamin K (90–120 μg), and beta-carotene (vitamin A, 5000 IU) [46,69].

According to the latest recommendations regarding pregnancy after bariatric surgery [46], patients should receive nutritional advice. In addition to standard postsurgical dietary advice, it is essential to individualize energy needs (based on pre-pregnancy BMI, gestational weight gain, and level of physical activity), consume protein at least 60 g/day, and limit rapidly absorbed carbohydrates. Nutrition should be systematically monitored before, during, and after pregnancy by checking specific serum indices every three months. Patients should be systematically screened for diabetes and have their body mass controlled.

Available studies evaluating the impact of the time from bariatric surgery to conception on obstetric outcomes assume similar time values for early pregnancy (≤12 months). They mostly agree that it is unfavorable for the pregnant woman and her offspring. Significant weight loss may result in nutrient deficiencies (including vitamin B12 and folic acid) and anemia [70], lower gestational weight gain, lower gestational age at delivery, lower birth weight, and preterm birth [71]. There was no relationship between gestational weight gain and the type of bariatric surgery [72].

However, some scientific reports do not support a correlation between the time from surgery to conception and early or late fetal growth in pregnancies conceived after gastric bypass surgery [73], as well as maternal outcomes (including pregnancy-induced hypertension and gestational diabetes) and neonatal outcomes independently on the type of operation [72]. Even studies report that pregnancies 18 months after gastric bypass, compared with pregnancies in the earlier period, are associated with a higher risk of cesarean section or intravenous iron supplementation [74]. There are also conflicting reports regarding the relationship between weight gain in pregnancies conceived at different times after bariatric surgery and low birth weight [72,74]. The latest recommendations do not specify the time after MBS to conception but indicate the moment of achieving stable body weight [46].

Systematic reviews of studies and meta-analyses on the impact of bariatric surgery on women’s health during pregnancy often do not differentiate the type of procedure performed. Conclusions from such studies and studies performed with the mixed method (Roux-Y-gastric bypass (RYGB)) are presented below. Possible health consequences of bariatric surgery for a pregnant woman include:-Deficiencies of vitamins and microelements, especially iron, vitamin B12, A, phylloquinone, and folic acid [75,76,77], as well as zinc, selenium, and vitamins A1, B1, B6, and C [78];-Hormonal changes—an increase in the level of sex hormone binding globulins (SHBG), progesterone, and estradiol and a decrease in the level of androgens, androstenedione, and testosterone; LH and FSH remained unchanged; studies on anti-Mullerian hormone (AMH) are inconsistent [55,77];-Improvement in the regularity of menstrual cycles and fertility [77];-Lower risk of gestational diabetes [9,12,76,79,80] but a higher risk of hypoglycemia and hyperglycemia [81];-Lower risk of gestational hypertension [9,12,76,79,80,82];-Higher risk of anemia [9,76];-Higher risk of fractures [9];-No effect on gestational weight gain [83].

Some scientific reports differentiate the effects of bariatric surgery depending on its type. On this basis, it is known that restrictive methods contribute to the increase in the level of folic acid, do not cause changes in the level of vitamin B12 and D, have no effect or increase the level of antimullerian hormone (AMH), do not affect the level of FSH and estradiol, have a positive effect on the regularity of menstrual cycles and fertility [77], and reduce the risk of pre-eclampsia [84]. In turn, methods that impair absorption cause an increase in the level of folic acid but a decrease in the level of vitamin B12, a decrease in the level of luteinizing hormone (LH) and estradiol, an increase in the level of follicle-stimulating hormone (FSH) and sex hormone binding globulin (SHBG), and an increase or decrease in testosterone and dehydroepiandrosterone sulfate; they do not change the level of androstenedione. The results of studies on the effect of the regularity of the menstrual cycle are inconsistent [77].

Although the vast majority of studies on the effects of bariatric surgery in women during pregnancy are based on the comparison of postoperative women with control groups with matched BMI (most often women with obesity), some reports refer to the general population [85] or women with a normal BMI [84]. Compared with these two groups, women during pregnancy who have undergone bariatric surgery are still more likely to have hypertension and gestational diabetes. At the same time, they may show changes in physiological patterns in the lipid profile, which make them comparable to women of normal weight (C-reactive protein) or lower weight (TC, LDL-C, and non-HDL-C) [86]. Hypoglycemia and large and rapid spikes in blood glucose levels are under-reported in women during pregnancy after bariatric surgery, and diagnosing gestational diabetes mellitus is more complicated. It is often based on the oral glucose tolerance test (OGTT), although it is considered unreliable in bariatric patients [87].

Regardless of the type of bariatric surgery, reports concerning women’s mental state during pregnancy are worrying. An increase in the incidence of mental disorders is reported, especially depression [88,89,90] and anxiety [89], as well as an increase in the incidence of self-harm and suicide [9], and a significant proportion of women who consume alcohol (33.5%) and use opioids (28.5%) have also been observed [90]. A significant relationship with depressive symptoms may include marital status, whether the pregnancy was planned and desired, and a history of mental disorders. In turn, mental disorders during pregnancy may increase the rates of abortion, hemorrhage, low birth weight, and negative consequences for the child’s psychosocial development. In addition, the psychiatric background may result in less adherence to prenatal care and difficulties in the woman’s acquisition of self-care habits, and it affects the quality of the mother–child relationship during pregnancy and postpartum, mainly due to its association with postpartum depression [88]. Considering the above facts, screening patients’ mental condition after MBS is recommended before, during, and after pregnancy. Monitoring for substance abuse, depression, and anxiety seems particularly important [46].

In addition to mental disorders, women after bariatric surgery experience many fears that negatively affect their mental well-being. Some of them coincide with typical concerns of pregnant women (e.g., whether the child will be born healthy). However, others are specific to this group of patients and concerned with the issues of proper nutrition and growth of the child, gastrointestinal problems (pain associated with dumping or bowel obstruction), weight gain and return to pre-surgery weight, and social judgment about eating too small portions and the negative impact of this on the baby. These concerns are independent of whether bariatric surgery was the goal for a consciously planned pregnancy or whether the pregnancy was unplanned or complicated [91]. There is also the issue of weight-related shame, self-blame, and fear of stigmatizing bariatric treatment in scientific reports. These factors affect women’s expectations regarding health care during pregnancy, follow-up visits, and the availability of information and may also be a barrier to seeking or accepting professional help. Lack of proper care and medical education during pregnancy negatively affects risk awareness, misconceptions about the impact of breastfeeding on weight loss, and the belief that weight gain during pregnancy is not a response to lifestyle changes. It reduces the motivation and tendency of women to lead a healthy lifestyle [92].

The bariatric surgery of a woman before pregnancy affects the course of pregnancy and obstetric outcome. Therefore, fetal ultrasound examinations should be performed routinely during pregnancy after MBS in the first two trimesters (12 and 20 weeks) and once monthly in the third trimester [46]. Scientific reports indicate the following effects of surgery, regardless of the type of surgery performed and in mixed procedures (RYGB):-Frequency of spontaneous abortion—inconsistent research results; some report no effect, while others report an increase in the frequency of miscarriages [77];-Fetal defects—research results are inconsistent [93]; some reports indicate no effect [77,94] or reduced risk [77,82]. However, some studies report an increased risk [9];-Preterm birth—results are mostly inconsistent [82], although a few reports report an increased risk [9,95,96];-Post-term delivery—lower risk [54,97];-Cesarean delivery—the results of reports are inconsistent [82]; some studies indicate a decrease in the risk of unplanned cesarean delivery [97];-Obstetric anal sphincter injury, postpartum hemorrhage—lower risk [97];-Transfer of a newborn to the intensive care unit (ICU)—higher risk [9];-Perinatal death—no effect [96] or increased [9,54];-Small for gestational age—higher risk [9,12,76,79,80,82,95,96];-Low birth weight—increased risk [54,83];-Adipose tissue—lower lean mass and percentage of fat [98];-Large for gestational age—a decrease in risk [9,80,96];-Fetal macrosomia—risk reduction [9,76,79,82];-Fetal consequences of vitamin deficiency—visual complications (vitamin A), intracranial hemorrhage (phylloquinone), neurological and developmental disabilities (vitamin B-12), and malformations (folic acid) [75].

The above studies compared women with obesity after bariatric surgery and women with obesity without surgical treatment, and studies concerned single pregnancies. However, few studies on twin pregnancies have also confirmed the association of bariatric surgery before pregnancy with delayed intrauterine growth of the fetus, as indicated by fetal parameters such as slower growth of subcutaneous fat tissue and abdominal circumference, as well as lower birth weight of the newborn [99].

Compared with the general population, women after bariatric surgery are at a higher risk of preterm delivery, labor induction, planned and unplanned cesarean sections, having a child small for gestational age, and having low birth weight [85]. Regarding the impact of MBS on pregnancies resulting from in vitro fertilization (IVF), it has been shown that women from this group, compared with women with the same BMI but not surgically treated, produce fewer oocytes and have fewer frozen embryos, and their children have a much lower birth weight. However, there was no association between MBS and a reduction in the cumulative live birth rate [100].

Compared with the population of women with normal weight, pregnant women after bariatric surgery are more likely to have acute abdominal pain during pregnancy; low birth weight; neonatal admission to the intensive care unit [84]; reduced (<2.5 percentile) concentration of calcium; and elevated (97.5th percentile) of magnesium, vitamin E, 25-hydroxyvitamin D, and vitamin B12 [101]. The woman’s body weight immediately before conception seems essential for the discussed results. Despite bariatric surgery, most women become pregnant with a BMI > 30, which may negatively impact obstetric outcomes [102]. On the other hand, in a 2023 systematic review of studies, no clear and consistent associations were identified between dietary intake, supplementation, or GWG and micronutrient deficiency [103]. Nutritional deficiencies can occur in a pregnant woman, regardless of her body weight or whether she has previously undergone bariatric surgery. In every case, nutritional deficiencies negatively influence fetal development, neonatal complications, and long-term outcomes [75,104,105]:-Vitamin B12—fetal malformations, anemia, neutropenia, delayed gross motor, and delayed speech;-Vitamin B9—birth neural tube defect, spinal cord or brain defect, and subsequent development of encephalocele; hydranencephaly; anencephaly; spina bifida, including meningocele and myelomeningocele;-Vitamin A—congenital abnormalities, premature birth, ventricular dilatation, retardation, bilateral microphthalmia, ocular malformations, and retinal damage;-Vitamin D—skeletal defects, inferior adherent leucoma, and light-sensitive child;-Vitamin K deficiency—optic nerve hypoplasia and musculoskeletal defects;-Folic acid—neural tube defects and ventricular septal defects;-Iron (Fe), calcium (Ca), and zinc (Zn)—preterm birth, epilepsy, severe anemia, blindness, and deafness.

The mental state affects the course of pregnancy, childbirth, and the postpartum period. In addition to the patient’s individual conditions, the relationship with the medical staff plays a significant role. In the case of women during pregnancy and those giving birth after bariatric surgery, in addition to typical relational needs (sense of security and support), the knowledge of medical staff related to bariatric surgery and the willingness to talk about it play an essential role [91]. Midwives’ empathy and awareness of the social stigma of obesity may paradoxically prevent them from discussing body weight for fear of embarrassing or worrying the patient [92]. This indicates the need for professional psychological education of medical staff in the area of interpersonal communication in general, with particular emphasis on the issues of obesity and bariatric surgery.

## 6. Conclusions and Perspectives of Further Research

Women with obesity planning pregnancy should reduce their body weight because obesity is a risk factor for adverse obstetric and neonatal outcomes. In turn, bariatric surgery effectively reduces excessive body weight and health risks in women with obesity during pregnancy and in their offspring (Table 3).

An interval of at least a year between surgery and conception is recommended, as well as the monitoring of nutritional status, especially the level of vitamins and minerals. An interdisciplinary medical team with knowledge and skills about people after bariatric surgery should care for a patient during pregnancy. Due to the increased risk of mental disorders, especially depression, it is necessary to monitor the woman’s mental state constantly and provide psychological support and education on a healthy lifestyle during pregnancy and postpartum. Figure 1 illustrates the main individual stages of professional medical care for a patient with obesity before and during pregnancy.

The postpartum period is critical for weight management to prevent the risk of long-term chronic diseases and prepare for another healthy pregnancy. However, postpartum weight retention (PPWR) may be supported by lack of sleep, fatigue, unfavorable eating habits, and insufficient physical activity. Therefore, in women with obesity before or during pregnancy, it is essential to systematically control body weight [107]. Similarly, in women after metabolic and bariatric surgery. In this group, pregnancy does not affect long-term weight loss. For example, in AGB management, a return band to pre-pregnancy levels is recommended after the establishment of lactation.

Women after MBS can breastfeed, but it is advisable to monitor maternal micronutrients. Patients with gestational diabetes mellitus should be offered screening for type 2 diabetes in the postpartum period [46]. An important issue for both groups of discussed patients is monitoring their mental condition regarding postpartum depression and other mental disorders [18,91].

Further research is needed to understand better the impact of obesity itself and weight loss after bariatric surgery on pregnancy and obstetric outcomes. The contradictory results in scientific reports regarding the birth weight of newborns of mothers with obesity require a better determination of the factors and mechanisms that determine the final result. It is necessary to supplement the research on the relationship between bariatric surgery’s positive physical and mental effects and the performance of abdominoplasty in patients as well as on the long-term impact on children of women who underwent bariatric surgery before pregnancy. As demonstrated in a review of studies by Kapadia et al. [27], the lack of a relationship between negative affect (anxiety, stress, and depression) and excessive weight gain during pregnancy may differentiate the population of women during pregnancy and the general population, which also requires a more thorough study. The number of scientific reports on the psychological aspects of pregnancy in women with obesity and after bariatric surgery is insufficient. The intensification of scientific research in this area may increase the competence of medical staff and thus positively impact the effectiveness of obstetric care for women with obesity.

## Figures and Tables

**Figure 1 nutrients-15-04289-f001:**
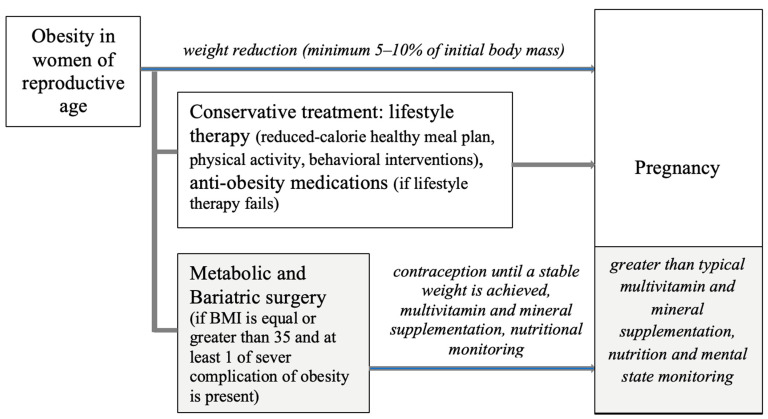
Pregnancy-related indications for women with obesity and after metabolic and bariatric surgery. Sources: [46,69,70,91,92,106].

**Table 1 nutrients-15-04289-t001:** Search terms used in the review.

Subject	Search Term
Epidemiology of obesity among women of reproductive age	obesity, epidemiology, women, procreation period
Obesity and pregnancy— a medical perspective	obesity, procreation, conception, pregnancy, birth postpartum period, complications, risk, consequences, obesity treatment, weight loss, obstetric outcome, neonatal outcome
Obesity and pregnancy— a psychological perspective	obesity, pregnancy, psychological state, emotional state, depression, media, pop culture, body image, consequences
Bariatric surgery in women of reproductive age— benefits and risks	bariatric surgery, benefit, risk, women, procreation period
The specificity of pregnancy in women after bariatric surgery— a medical perspective	pregnancy after bariatric surgery, nutrition, malnutrition, complications, risk, consequences, obstetric outcome, neonatal outcome
The specificity of pregnancy in women after bariatric surgery— a psychological perspective	pregnancy after bariatric surgery, body image, psychological state, emotional state, social support, quality of life

**Table 2 nutrients-15-04289-t002:** Weight gain recommendation for women in single pregnancy.

Weight Status before Pregnancy	Weight Gain during Pregnancy
Overweight (BMI 25.0–29.9 kg/m^2^)	from 2 to less than 16 kg
Class 1 obesity (BMI 30–34.9 kg/m^2^)	from 2 to less than 6 kg
Class 2 obesity (BMI 35–39.9 kg/m^2^)	from 0 to less than 4 kg
Class 3 obesity (BMI ≥ 40 kg/m^2^)	from 0 to less than 6 kg

Source: [13].

**Table 3 nutrients-15-04289-t003:** The impact of obesity and bariatric surgery on pregnancy and perinatal outcomes.

	Effect of Obesity on Pregnancy (Women with Obesity vs. Women with Normal Body Weight)	Effect of Bariatric Surgery on Pregnancy (Women with Bariatric Surgery vs. Women without Surgery with the Same Body Weight)
Mother	Gestational diabetes mellitus (GDM)		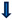
Gestational hypertension (GH)		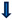
Pre-eclampsia (PE)		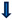
Vitamin and mineral levels	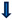 (vit. D)	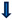 (iron, phylloquinone, folic acid, zinc, selenium, vit. A1, B1, B6, B12, C)
Mental state	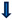 (anxiety, depression, postpartum depression, binge eating disorders, serious mental illness)	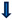 (anxiety, depression, self-harm, suicide)
Other	 (perinatal cardiomyopathy, obstructive sleep apnea, thromboembolism, stroke, myocardial infarction, wound infections)	 (anemia, fractures, menstrual cycle regularity, fertility)changes in hormone levels
Child	Spontaneous abortion (SAB)		 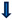
Fetal defects	 (neural tube defects, hydrocephalus, heart, craniofacial and limb defects)	 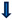
Preterm birth		
Cesarean delivery (CD)		 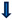
Low birth weight (LBW)		
Small for gestational age (SGA)		
Large for gestational age (LGA)		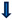
Fetal macrosomia (FM)		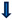
Perinatal death/stillbirth		
Postpartum complications	 (strenuous and prolonged delivery, postpartum hemorrhage, venous thromboembolism, reduced frequency of breastfeeding initiation and its shorter duration, complications resulting from protracted immobilization)	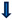 (obstetric anal sphincter injury, postpartum hemorrhage)  (stay of a newborn in the Intensive Care Unit)
Negative consequences in the future	 (overweight and obesity, ischemic heart disease, stroke, type 2 diabetes, asthma, potentially also immunological and infectious diseases, worse cognitive functions, increased risk of neurodevelopmental disorders, including cerebral palsy)	? unknown


—risk increase; 
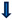
—risk decrease; 


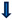
—inconsistent results. Source: Own study based on [9,10,11,12,13,14,15,16,25,54,75,76,77,78,79,80,81,82,83,84,88,89,90,93,95,96,97].

## Data Availability

No new data were created or analyzed in this study. Data sharing is not applicable to this article.

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
