# Peer review of "Medical and Psychological Aspects of Pregnancy in Women with Obesity and after Bariatric Surgery"

_nutrients, 2023, doi:10.3390/nu15194289_

Round 1
Reviewer 1 Report
The topic addressed is of interest but the methodology chosen detracts from the value of the study.
A narrative review is a deficient technique because it does not explain the search strategy exhaustively and without a concrete method, nor does it clearly describe how the studies were selected.
The analysis of the results is not critical and rigorous, nor is it based on a protocol, but is a simple description of the study findings.
The synthesis of information is not based on scientific methodology, nor is it based on data extraction and guidelines such as PRISMA, but is based on a subjective methodology.
The interpretation of results does not usually discuss sources of error (sensitivity analysis, publication bias).
I think it would have been more interesting for the readers to perform a systematic review and not a narrative.
Author Response
Dear Reviewer, thank you for your comment. We agree that a systematic review has much greater scientific value than a narrative one. In our case, the latter option was primarily due to limited capabilities due to the number of team members and finances. However, we decided to take up the topic because we did not find a published article in the scientific databases that broadly describes both the medical and psychological aspects of obesity before and during pregnancy. Meanwhile, we consider this topic crucial both scientifically and for clinical practice. The medical community (physicians, dietitians, physiotherapists, psychologists) caring for women with obesity before and during pregnancy needs knowledge of how to do it professionally and effectively. Our article can contribute to this. It contains information from peer-reviewed scientific journals, which may guide researchers in deepening and specifying these issues and for clinicians to observe the described relationships in practice. We carried out the methodological process to the best of our knowledge and care and described its details in the manuscript. Although narrative reviews have their limitations, they are published in peer-reviewed scientific journals, including Nutrients:
- Guo J, Lovegrove JA, Givens DI. A Narrative Review of The Role of Foods as Dietary Sources of Vitamin D of Ethnic Minority Populations with Darker Skin: The Underestimated Challenge. Nutrients. 2019; 11(1):81. https://doi.org/10.3390/nu11010081
- Petroski W, Minich DM. Is There Such a Thing as "Anti-Nutrients"? A Narrative Review of Perceived Problematic Plant Compounds. Nutrients. 2020 Sep 24;12(10):2929. doi: 10.3390/nu12102929. PMID: 32987890; PMCID: PMC7600777.
- Grant WB, Al Anouti F, Boucher BJ, Dursun E, Gezen-Ak D, Jude EB, Karonova T, Pludowski P. A Narrative Review of the Evidence for Variations in Serum 25-Hydroxyvitamin D Concentration Thresholds for Optimal Health. Nutrients. 2022; 14(3):639. https://doi.org/10.3390/nu14030639
Reviewer 2 Report
Dear author,
Thank you for sharing your research. After careful revision, I have some questions and suggestions for improvement:
1. Please correct the title number 4, which is currently in your native language.
2. I found the role of weight loss before pregnancy and bariatric surgery (both separately and in the same patient) to be interesting. It would be helpful if you could discuss the potential impact of these factors on the post-pregnancy period in more detail.
3. Furthermore, it seems that weight loss during pregnancy is considered unhealthy, while moderate weight gain is recommended. On the other hand, weight loss during the pre-pregnancy period is favorable, especially if followed by bariatric surgery in patients who need it. However, in this case, a pregnancy should not be planned within one year due to potential endocrinological disorders that could affect both the mother and the baby. Could you prepare a timeline to illustrate the different planning involved?
It may also be beneficial to include another table that outlines the recommended weight gain during pregnancy based on BMI.
Thank you for considering these suggestions.
Author Response
Dear Reviewer,
Thank you very much for your valuable review. We have taken into account all your suggestions. Due to the review, the quality of the manuscript has improved significantly. You can find the detailed response to your comments in the attachement. All changes in the manuscript have been marked in red.

Reviewer 3 Report
1.The summary is written unstructured and disorganized. The authors do not follow the criteria for the preparation of the abstract.
2.The authors completely wrongly define the factors on the basis of which we evaluate the nutritional status; BMI is not a criterion for assessing nutritional status. US is not a suitable method for assessing obesity and nutritional status.
3. The ahthors should clearly define the term wound infection.
4. line 122: written unintelligibly.
5. Can we define reduced frequency and duration of breastfeeding, as a plot?
6. The authoirs shoud clearly present - complications from prolonged immobilization: reasons, systemic relationship in clinicl practice?
7. line 139: the authors must precisely define the type of deficiency that can affect the development of the fetus and define according to the type of nutrition of pregnant women in relation to congenital disorders.
8. line 144-159: needs further explanation and clear evidence. It is written in general without description of any correlation.
9. line 235 written / not translated
10. line 293-306: many mistakes written regarding the technic, physiology and nutritional consequences. Not accaptable.
11.
11. The authors must define precisely the recommendations for patients after bariatric surgery, who are either preparing for pregnancy or without it, and take into account the recommendations for follow-up after surgery and nutritional treatment. They should define weight gain after bariatric surgery and the causes, as well as the state of mental health.
12: line 487- 496: shoud be clearly written and explained; authors should coment on guidelines for pregnany after BS
English should be re-edited.
Author Response
Dear Reviewer,
Thank you very much for your valuable review. We have taken into account all your suggestions. Due to the review, the quality of the manuscript has improved significantly. You can find the detailed response to your comments in the attachment. All changes in the manuscript have been marked in red.

Round 2
Reviewer 1 Report
Congratulations for the work done. I encourage you to continue in this line of work.